# Post-Ischemic Permeability of the Blood–Brain Barrier to Amyloid and Platelets as a Factor in the Maturation of Alzheimer’s Disease-Type Brain Neurodegeneration

**DOI:** 10.3390/ijms241310739

**Published:** 2023-06-27

**Authors:** Ryszard Pluta, Barbara Miziak, Stanisław J. Czuczwar

**Affiliations:** Department of Pathophysiology, Medical University of Lublin, 20-059 Lublin, Poland

**Keywords:** brain ischemia, Alzheimer’s disease, blood–brain barrier, platelets, pericytes, astrocytes, amyloid, tau protein, amyloid plaques, neurofibrillary tangle, microbleeding, neurodegeneration

## Abstract

The aim of this review is to present evidence of the impact of ischemic changes in the blood–brain barrier on the maturation of post-ischemic brain neurodegeneration with features of Alzheimer’s disease. Understanding the processes involved in the permeability of the post-ischemic blood–brain barrier during recirculation will provide clinically relevant knowledge regarding the neuropathological changes that ultimately lead to dementia of the Alzheimer’s disease type. In this review, we try to distinguish between primary and secondary neuropathological processes during and after ischemia. Therefore, we can observe two hit stages that contribute to Alzheimer’s disease development. The onset of ischemic brain pathology includes primary ischemic neuronal damage and death followed by the ischemic injury of the blood–brain barrier with serum leakage of amyloid into the brain tissue, leading to increased ischemic neuronal susceptibility to amyloid neurotoxicity, culminating in the formation of amyloid plaques and ending in full-blown dementia of the Alzheimer’s disease type.

## 1. Introduction

We are currently observing an increased interest in research on the behavior of the blood–brain barrier (BBB), including the barrier after cerebral ischemia in the context of the development of neurodegenerative diseases and the possibilities of their prevention or treatment. Understanding the mechanisms of damage to the BBB during ischemia and recirculation may provide interesting clues related to neuropathological mechanisms that are important in clinical practice, including post-ischemic dementia and Alzheimer’s disease (AD) [1]. BBB dysfunction complicates cerebral ischemia [2,3,4,5,6,7,8] and AD [9,10,11,12,13,14,15]. In a transgenic model of AD, insufficiency of the BBB has been observed to precede the development of amyloid plaques and the clinical manifestation of the disease [16]. On the other hand, the accumulation of β-amyloid peptide has been shown to cause the death of endothelial cells [17], which are an important part of the BBB. Insufficiency of the BBB results in hyperphosphorylation of tau protein and conversely, pathological changes in tau protein cause damage to the BBB [18,19,20]. The consequences of permeability of the BBB can lead to the leakage of neurotoxic molecules from the blood, such as β-amyloid peptide and tau protein, into the brain parenchyma, resulting in neuronal death, amyloid plaque and neurofibrillary tangle formation and dementia [6,12,14,19,21,22,23,24,25,26,27,28,29,30,31,32,33,34,35]. Some evidence indicates that about 80% of the amyloid plaques in the transgenic model of AD [36] and about 90%of human amyloid plaques are in close contact with BBB vessels [37]. Therefore, we suggest that post-ischemic failure of the BBB is a factor in the maturation of lesions after ischemic brain injury, which ultimately leads to the development of post-ischemic dementia. In this review, we will show that the post-ischemic BBB behaves similarly to the BBB in AD.

## 2. Search Strategy and Selection Criteria

The articles cited in this review were searched in PubMed, Google Scholar, and Web of Science databases with key search terms including cerebral ischemia, stroke, BBB, brain endothelial cell, pericytes, astrocytes, β-amyloid peptide, amyloid, tau protein, and AD. The search mainly included works published in the years 2000–2023. Previous original papers of the first scientific discovery are also cited.

## 3. Structure and Functions of the BBB

Neurons are arranged in distinctive networks and structures. The environment of neuronal cells is tightly regulated, and any harmful elements must be removed. To this end, the brain has protective mechanisms that separate it from the rest of the body. In addition to structures and functional networks, there is another functional unit in the brain called the neurobarrier [38]. The neurobarrier consists of four different barriers, namely the neuronal and glial membrane barrier, the cerebrospinal fluid-ependyma barrier, the blood-cerebrospinal fluid barrier, and finally the classic BBB. Under physiological conditions, the BBB is impermeable to pathogens. Low vesicular transport by endothelial cells is responsible for maintaining the physiological function of the BBB [39]. The BBB plays an important role in protecting the brain tissue against the uncontrolled influx of toxic substances. Under physiological conditions, the BBB is a diffusion BBB, important for the proper functioning of the central nervous system. In addition, some agents, for instance, oxygen and carbon dioxide, freely diffuse through endothelial cells according to their concentration gradient [40]. Amino acids and glucose pass the BBB by transporters, while larger molecules such as insulin and leptin cross the BBB via receptor-mediated endocytosis [38,41]. In contrast, in various types of neuropathology, many molecules are released in the brain, such as glutamate and aspartate, which cause acute and/or chronic dysfunction of the BBB [2,42,43].

The main structural element of the BBB is the endothelium. In addition to the endothelial cells, the barrier consists of a continuous basement membrane, pericytes embedded in the basement membrane, and astrocytes covering the microvessel from the outside with its own endfeet [44]. In addition, tight junctions between brain microvascular endothelial cells have been documented. The tight junctions and endothelial cells together form the continuous structure of the BBB. The BBB is the gatekeeper between the circulating elements of the blood and the brain tissue. The length of the microvessels of the BBB in the adult brain is about 640 kilometers [45], the total area is about 12 m^2^ and the diameter is 0.3–0.5 µm [46]. Eighty-five percent of the cerebral vessels are made up of capillaries that contain the BBB, which is composed of, among other things, monolayer endothelial cells sealed with tight junctions to ensure low transcellular and paracellular transmission [47]. These vessels are surrounded by a specialized basement membrane, and pericytes in the same basement membrane contribute to the formation of the BBB in the embryo and its maintenance in adulthood. Pericytes are important in terms of the formation and maintenance of tight junctions at the BBB and the control of endothelial transcytosis. The tight junction proteins that make up the BBB are mainly: claudin-1, -3, -5, and -12 and occludin which are attached to the intracellular scaffold via zonula occludens-1, 2, 3. There are studies showing that the loss of pericytes results in the loss of claudin-5, occludin, and zonula occludens-1 or their rearrangement [48,49]. Pericytes have been shown not only to affect tight junctions but also to increase the permeability of the BBB by controlling endothelial passage [50]. The BBB is physiologically equipped with specific transport systems on the side facing the blood and brain. On the luminal side, nutrient transporters and regulatory molecule receptors control the influx of blood elements into the brain parenchyma through the endothelium. On the other hand, reverse transport on the abluminal side removes waste products from the brain parenchyma into circulation [44,51].

Crosstalk between the basement membrane and pericytes, especially astrocytes in the brain forming the BBB, provides contact between blood vessels and neuronal circuits, which causes them to deliver nutrients to brain tissue through this connection [52]. In addition, the water balance and extracellular ions in the BBB are controlled by the water channels and ions that are found in the endfeet of astrocytes [53].

The basal lamina is a layer of extracellular matrix known as the basal membrane, which consists of collagen, laminin, and fibronectin. Astrocytes are found around the cerebral microvessels and control the function of the BBB through astrocyte-derived factors and astrocyte terminal processes called endfeet. The potassium channel, Kir4.1, and aquaporin-4 are located in the endfeet of astrocytes and support BBB function by controlling ionic and water balance to prevent cerebral edema [14,54].

Moreover, in the BBB, astrocytes are converted from a quiescent to a reactive form after injury, and several astrocyte-derived factors induce endothelial cell apoptosis and down-regulate endothelial tight junction proteins, leading to impaired BBB [53]. On top of it, some factors derived from astrocytes may protect endothelial cells and enhance tight junction reassembly, leading to the reconstruction of the BBB [34,55]. In addition, several astrocyte-derived factors also regulate cell adhesion molecules in endothelium and control leukocyte passing [53].

## 4. BBB Permeability Post-Ischemia

The impact of post-ischemic brain damage on the function of the BBB is currently the subject of intensive research, among others, in the context of preventing or treating neurodegenerative changes with the use of substances that would pass through the barrier to the damaged brain tissue. An ischemia-reperfusion episode causes a series of changes that increase the permeability of the BBB to cellular and non-cellular blood components, lead to the opening of tight junctions, and sometimes to diffuse leakage of all blood elements through the necrotic vessel wall [6,34,36,56,57,58,59,60,61,62,63]. In ischemia-reperfusion injury of the BBB, two abnormal and characteristic features deserve attention. One is important given the chronic effects of extravasated substances, such as the neurotoxic β-amyloid peptide, in generating neurodegenerative irreversible neuropathology, and the other concerns the leakage of cellular elements of the blood e.g., platelets, resulting in acute, massive, and mechanical destruction of brain parenchyma [6,64,65]. On the other hand, cells of peripheral tissues and organs are known to continuously produce the neurotoxic β-amyloid peptide [66]. The ability of **the** β-amyloid peptide to cross the damaged BBB may lead to local neurotoxic effects on certain neuronal cell populations, including increased production and accumulation of β-amyloid peptide in the brain parenchyma [27,30]. Circulating β-amyloid peptide can be delivered to ischemic brain parenchyma and its microcirculation, and thus may contribute to brain amyloidosis after an ischemia-reperfusion episode in stroke patients [27,30,67,68,69,70,71,72,73].

### 4.1. Permeability of Non-Cellular Blood Elements through the Ischemic BBB in the Gray Matter

One year after transient cerebral ischemia in rats, brain slices demonstrated multifocal areas of extravasated horseradish peroxidase in gray matter used to assess the permeability of the BBB [6,32,33,34,35,56,57]. Light microscopic examination of vibratome brain sections revealed many diffuse and focal staining in the cortical layers of horseradish peroxidase. Many penetrating blood vessels also showed a reaction to horseradish peroxidase of the vessel walls. Horseradish peroxidase was seen in endothelial cells and outside the vessels. In other brain structures, such as the hippocampus, thalamus, basal ganglia, and cerebellum, diffuse as well as isolated multiple extravasation sites of horseradish peroxidase were found. The permeability of the BBB post-ischemia was not restricted to a specific gray matter brain structure, but was mainly dominated by the branching and bifurcation of blood vessels [6]. Overall, following cerebral ischemia, animals exhibited random and focal changes in gray matter in the BBB. Extravasations of horseradish peroxidase were localized in the perivascular space of microvessels, arterioles, and venules. Extravasations of horseradish peroxidase around the leaking vessels resembled “puffs of smoke”. The above changes in the ischemic BBB were accompanied by atrophy of the brain cortex and especially of the hippocampus [31,74,75].

Human β-amyloid peptide 1–42 was found after intravenous injection in the vascular walls and perivascular space in rat post-ischemic cortex with a survival of 3 months [27,28,30,51]. It should be noted that the β-amyloid peptide alone can cause dysfunction of BBB by disrupting endothelial functions and/or endothelial cell death [76,77,78].

Six months post-ischemia, animals showed increased perivascular immunoreactivity in gray matter for all parts of the amyloid protein precursor [9,74]. At survival times greater than 6 months, staining of only the β-amyloid peptide and C-terminal of amyloid protein precursor has been noted [31,32,33,34,35,79]. Staining of different parts of the precursor was mainly observed in the extracellular space in gray matter such as the cortex and hippocampus. Numerous extracellular accumulations of C-terminal of amyloid protein precursor and β-amyloid peptide adhered to or mainly embraced capillaries, spreading multifocally in gray matter. The accumulations had an irregular shape and were of various sizes and very well outlined.

The perivascular fragments of the amyloid protein precursor that surrounded the cerebral vessels formed perivascular cuffs or “puff of smoke”-like areas. In addition, the vascular lumen and pericytes and the inner and outer sides of the capillary walls accumulated fragments of the amyloid protein precursor. Accumulation of amyloid and C-terminal of amyloid protein precursor around cortex vessels indicates diffusion of the C-terminal of amyloid protein precursor and β-amyloid peptide from the microcirculatory compartment [27,30,34]. Strong perivascular and vascular amyloid accumulation has been demonstrated in the entorhinal cortex, hippocampus, and brain cortex.

### 4.2. Permeability of Non-Cellular Blood Elements through the Ischemic BBB in the White Matter

Post-ischemia BBB in the white matter showed progressive and chronic insufficiency [35,36,62]. Micro BBB changes predominated in periventricular and subcortical white matter and were random and spotty [35,36,62,80]. Extravasation of horseradish peroxidase was observed around the capillaries, arterioles, and venules [36]. Damaged endothelial cells and pericytes filled with horseradish peroxidase were less observed than in gray matter [6,56,57]. Perivascular immunoreactivity to all amyloid protein precursor fragments was evident in rats’ white matter up to 6 months post-ischemia [31,79]. After cerebral ischemia-reperfusion with a survival of >6 months, both the toxic C-terminal of the amyloid protein precursor and β-amyloid peptide around the BBB vessels, developing perivascular cuffs with rarefaction of the adjacent white matter and parallel oligodendrocyte staining were noted [31,32,33,34,35,36]. Accumulation of the C-terminal fragment of amyloid protein precursor and β-amyloid peptide dominated in the corpus callosum, subcortical region, and around the lateral ventricles [36,80]. These observations of BBB permeability were confirmed after intravenous administration of human β-amyloid peptide 1–42 after cerebral ischemia in a rat [27,28,29,30].

### 4.3. Permeability of Cellular Blood Elements through the Ischemic BBB in the Gray Matter

Platelet aggregation in cortical blood vessels has been observed for 1 year post-ischemia [10,32,34,75]. As a result of these changes, there were several vessels partially or completely blocked by platelets [10,32] and/or platelets with their membranous remnants [56]. Platelets were also visualized outside the microvessels in gray matter [10,32,34]. In the areas already presented, the endfeet of the astrocytes were heavily swollen [32]. Platelets in the vascular lumen dominated in capillaries and venules. The platelets usually had well-developed pseudopodia, which in many cases were in direct contact with the endothelium [10]. In addition, the projection of endothelial microvilli was directed toward the platelets in the lumen of the vessels [4]. The presented changes occurred in arterioles, venules, and capillaries, regardless of survival time after brain ischemia. In contrast, some data suggest that cerebral ischemia triggers the creation of platelet and leukocyte aggregates, which often interact with endothelial cells [81,82]. Many years of research indicate that leukocytes play a key role in cerebral ischemic episodes [64,65,83,84,85,86,87]. It is believed that leukocytes with platelets block microcirculation, which promotes the development of hypoperfusion and no-reflow phenomenon after cerebral ischemia [88]. Leukocytes cause pathological changes in neurons through the release and interaction of different types of inflammatory molecules [65,89,90]. Some data suggest that leukocytes, most likely neutrophils, are the key cellular source of matrix metalloproteinase-9 after cerebral ischemia [87]. Neutrophil matrix metalloproteinase-9 recruited to ischemic brain gray matter promotes further recruitment of neutrophils to the same areas of the brain in a positive feedback fashion and causes secondary alterations to the BBB [65]. Thus, neutrophil-derived matrix metalloproteinase-9 directly contributes to post-ischemic brain damage [87]. Studies of the BBB using an electron microscope allowed the identification of polymorphonuclear and mononuclear leukocytes adhering to the endothelial cells of capillaries and venules from the lumen side [86]. Observations of the projection of pseudopodia of leukocytes and endothelium facing each other indicate the attachment and adhesion of endothelial cells to white blood cells [86]. It is assumed that this phenomenon probably plays an important role in the passage of white blood cells through the endothelium. Leukocytes may reduce local cerebral blood flow by constricting and/or blocking cerebral blood vessels [58,91]. Increased neutrophil endothelial adhesion mediators and cytokines promote the migration of white blood cells across the ischemic BBB [92]. In this way, the recruitment of white blood cells appears to activate molecular mechanisms that lead to endothelial tight junction disruption, BBB insufficiency, and ultimately progressive brain gray matter damage with microbleeding [24,92,93,94,95,96].

### 4.4. Permeability of Cellular Blood Elements through the Ischemic BBB in the White Matter

Electron microscopy studies after cerebral ischemia-reperfusion injury with survival of up to 1 year have shown single platelet aggregates in and out of capillaries, venules, and arterioles in the white matter [32]. The platelets inside and outside the cerebral vessels were irregularly shaped and had numerous pseudopodia. Platelets were often attached to leaky microvascular endothelial cells. Single vessels were completely blocked by aggregating platelets and their membranous remnants [32,34].

Platelet aggregation along with red and white blood cells caused microblading and complete microcirculation occlusion, resulting in local areas without recirculation after cerebral ischemia [10,24,32,34,75,81,82,94,95,96,97]. The no-reflow phenomenon [88] persisted all the time after the resumption of circulation in the brain following focal ischemia and caused a systematic increase in infarct volume [32,97]. These observations confirm the important role of blood cells in neuropathology during acute and chronic periods of recirculation and their negative impact on the neurological outcome after ischemia with reperfusion.

## 5. Platelets as Source of Amyloid Protein Precursor and Amyloid

A soluble form of the β-amyloid peptide is present in plasma after cerebral ischemia [70,72,73]. Platelets are known to be the major source of both amyloid protein precursor and β-amyloid peptide in post-ischemic blood. More than 90% of the β-amyloid peptide present in the peripheral circulation has been shown to be derived from platelet α-granules. The age of the circulating platelets affects the level of β-amyloid peptide and amyloid protein precursor in the blood following brain ischemia. Platelets under normal conditions live 7–11 days. β-amyloid peptide has been shown to play an important role in the proper functioning of platelets and coagulation mechanisms [45]. Since β-amyloid peptide causes vasoconstriction [58,98] and damage to endothelial cells [99], it has been suggested that β-amyloid peptide is actively involved in ischemia-reperfusion injury of brain tissue.

Chronic ischemic damage of the BBB [32,33,34,35,36] and platelet aggregates in the perivascular space [10,32] together with neurotoxic deposits of amyloid protein precursor fragments may be involved in the gradual maturation of neuropathology in ischemic gray and white matter leading to progressive post-ischemic dementia during survival [12,100,101,102,103,104]. Progressive damage to the gray and white matter after cerebral ischemia may be caused not only by the degeneration of neurons during an ischemic episode but also by neuropathological changes in the BBB with a large permeability of neurotoxic fragments of the amyloid protein precursor [6,62,74,105,106]. This accumulation can lead to retrograde neuronal death and pathological tau protein changes in oligodendrocytes [107]. There is an alternative theory suggesting that a silent and repetitive micro-ischemic episode may underlie the development of progressive neurodegeneration with post-ischemic dementia with the AD phenotype (Figure 1). The profile of pathological changes in the gray and white matter observed after cerebral ischemia shares features with changes in the brains of patients with AD. On the other hand, periventricular changes in the white matter after cerebral ischemia and in AD are referred to as leukoaraiosis [11,35,36,80]. All types of damage presented are responsible for behavior, cognition, and the general functioning of the body.

## 6. Amyloid and BBB Vascular Angiogenesis

Post-ischemic angiogenesis is important for the repair of brain gray and white matter, i.e., the repair and healing of damaged areas and structures, and for the prevention of recurrent ischemic episodes (Figure 1). Thus, an imbalance in angiogenesis adversely affects neurological outcomes. Islets of necrotic endothelial cells were observed in the BBB microvessels during recirculation, indicating severe damage to the BBB post-ischemia [6,57]. The necrotic BBB is characterized by a diffuse and massive leakage of pathogenic particles into the brain gray and white matter [6,57]. This phenomenon is probably due to the accelerated aging of endothelial cells. There is evidence in the literature supporting endothelial cell senescence in vivo [108]. Senescent endothelial cells are a hallmark of vascular aging [108,109,110] and are likely to be accelerated by cerebral ischemic events [57]. According to limited data, cerebral vessels are constantly modified during aging and brain diseases such as ischemic injury [111]. The induction of new capillaries by endothelial cells has been shown to be limited in an animal model of AD [77]. Deposition of the β-amyloid peptide in the vessels, as well as the peptide itself, are also anti-angiogenic factors [78]. In addition, observations from in vitro studies have shown that the β-amyloid peptide 1–42 is neurotoxic, suggesting that it plays a key role in the development of cerebrovascular pathology in AD [112]. Post-ischemic and β-amyloid peptide-dependent changes in pericytes and astrocytes affect angiogenesis, which is associated with normal BBB activity [34,55]. It should be added that some of the molecular mechanisms that control angiogenesis may also regulate neurogenesis in the brain parenchyma [111,113]. It is supported by a study that presented the strong interaction of angiogenesis and neurogenesis in the mature brain [114]. In addition to ischemic neuronal cell changes, damage to the BBB and problems with its repair [42,45], vascular regression, and inappropriate and abnormal angiogenesis may represent novel pathogenetic processes associated with the progression of neurodegenerative diseases and formation of amyloid plaques that are secondary to damage and/or death of ischemic neuronal cells [115]. Thus, it is likely that aging of the cerebral microcirculation following ischemia [6,57] with impaired angiogenesis severely affects the response of endothelial cells to various conditions, which subsequently in turn affects the function of the BBB.

Islets degenerated by vascular ischemia of the BBB may act as seeds for amyloid plaques and mediate neuroinflammatory processes [6,57]. Thus, these observations support the notion that progressive ischemic failure of the BBB can cause microvascular changes in the brain, such as angiopathy in the pial and intracerebral arteries, vascular aging, and inflammatory response, all being found to occur in AD brains. Thus, impaired angiogenesis as a result of ischemic cerebrovascular injury may be both a cause-and-effect mediator in the neuropathogenesis of AD [116].

## 7. Amyloid, Platelets, and Post-Ischemic Vasoconstriction

A combination of cerebral microvessel examinations in electron and scanning microscopy post-ischemia showed vasoconstriction of brain vessels, i.e., transverse and longitudinal ridging on the surface of endothelial cells [13,58,98,117,118]. It was found that the observed changes in cerebral vasoconstriction after ischemia differ depending on the arrangement of muscles in the blood vessel wall [58]. Thicker arteries and arterioles narrow circumferentially due to the circumferential arrangement of muscles, and thinner-walled veins and venules constrict by shortening their length due to the longitudinal arrangement of muscles [58]. Many different biochemical substances are known to cause cerebral vasoconstriction including amyloid [98,99]. Nevertheless, the exact mechanisms are not fully understood, because the vasoconstrictive response is multifactorial and of great clinical importance. The vasoconstriction was characterized by massive folding of the endothelium and basement membrane, constriction of pericytes, and increase microvilli on the endothelial cells surface and contracting platelets activity [10,13,82,119]. As a consequence of the above changes, aggregating platelets form microthrombi attached to damaged endothelial cells, causing a continuous influx of constrictive substances such as thromboxane A2, β-amyloid peptide, serotonin, etc. [98,119,120]. Scanning microscopy of the endothelium showed enlarged junctional ridging, an increased number of microvilli of endothelial cells, and deep craters located mainly in parajunctional regions. Numerous and elongated endothelial microvilli probably serve to slow down white blood cells in particular as they roll over the surface of the endothelium just before attachment to it [121]. Microvilli are present on the surface of endothelial cells in specific cerebral blood vessels, i.e., in arterial but mainly in venous branch areas. The post-ischemic vasoconstriction study provides important data on, among other things, the direct contact of the endothelium with leukocytes and platelets [10,64]. The above ischemia-induced relationship can lead to serious consequences for the patient, e.g., the development of recurrent cerebral ischemia. Some data suggest that platelets play a key role in ischemic brain injury not only through aggregation, thrombus development, and as a source of β-amyloid peptide, but also through participation in inflammatory reactions with leukocytes [122]. In addition, studies have shown that platelet-derived P-selectin plays an important role in vivo in the recruitment of white blood cells to the brain tissue after an ischemia-reperfusion episode and has a significant impact on leukocyte-dependent tissue changes [64,122]. Platelets are an important source of P-selectin and β-amyloid peptide and both substances are located in α-granules and are released together into the systemic, brain gray and white matter circulation after pathological stimulus. Since β-amyloid peptide causes vasoconstriction in vivo [98,99], a role for β-amyloid peptide in the development of brain hypoperfusion has been proposed. The above observations partly explain the neuropathological role of cerebral amyloid angiopathy [12,104,123] in reducing cerebral blood flow while β-amyloid peptide accumulates in the ischemic cerebral vasculature. β-amyloid peptide has been shown to lead to prolonged vasoconstriction by reducing endothelial nitric oxide production and cerebral blood flow [98,124,125]. The β-amyloid peptide also interferes with transcapillary glucose transport [126]. Under these conditions, the neurons after cerebral ischemic injury will additionally suffer from hypoperfusion and malnutrition during recirculation, which makes the neuronal cells more sensitive to the direct and/or indirect neurotoxic effects of β-amyloid peptide. In conclusion, the aggregation and adhesion of platelets along the inner and outer walls of cerebral vessels during recirculation probably promotes the interaction of white blood cells with the endothelium and develops vasoconstriction, and ultimately is involved in the ischemic injury of the brain gray and white matter after recirculation. There is a high probability that β-amyloid peptide-mediated vasoconstriction is a key factor in recurrent and repeated episodes of transient ischemia after an initial ischemic injury. These observations suggest that during recirculation, the brain tissue can be saved from irreversible ischemic changes and the increased volume of ischemic infarction can be reduced. This effect can be achieved by a preventive strategy inhibiting the development of prolonged vasoconstriction as partial or complete vascular occlusion, which triggers permanent permeability of the BBB, resulting in vascular microbleeding and/or severe hemorrhagic brain damage with massive edema [10,14,24,32,33,34,35,36,37,80,94,95,96].

## 8. Cerebral Amyloid Angiopathy Post-Ischemia

Cerebral amyloid angiopathy is a convenient term describing the accumulation of amyloid in the walls of cortical and meningeal arteries, capillaries, arterioles, and veins. It should be noted that a transgenic model of AD with high concentrations of circulating β-amyloid peptide in the blood did not cause cerebral amyloid angiopathy [127]. This finding indirectly indicates that the translocation and deposition of the β-amyloid peptide in the cerebral vascular wall requires alterations in the permeability of the BBB [42]. Endothelium, pericytes, and astrocytes are involved in the regulation of the BBB; therefore, their damage and/or death, especially in microvessels, leads to focal permeability or necrosis of endothelial cells, which results in failure of the BBB [57]. Such changes in the BBB can trigger focal and/or complete cerebral ischemia, one of the risk factors for the development of AD [18,19,20,21,37,38,39,41,74,128,129]. Thus, accidental and persistent ischemic permeability of the BBB may start a continuous process of accumulation of circulating β-amyloid peptide in the wall of the cerebral blood vessels [27,70,72,73]. This phenomenon is called cerebral amyloid angiopathy [12,104,123]. Collagen accumulation and basement membrane thickening after cerebral ischemia influences the deposition of β-amyloid peptide in the vascular wall [32]. Moreover, the initial accumulation of β-amyloid peptide in the vascular wall after ischemia may cause degeneration of endothelial cells and pericytes, which affects the activity of the BBB. In turn, dysfunction of the BBB leads to the incorporation of the β-amyloid peptide from the circulating blood and interstitial fluid, which is responsible for further accumulation of β-amyloid peptide and ultimately irreversible vascular degeneration. Under these conditions, the β-amyloid peptide from the plasma and interstitial fluid can also interact with the inner and outer parts of the capillary wall. Because of the large vascular network in the brain, it seems likely that amyloid plaques are associated with cerebral ischemic vessels. Extensive accumulation of β-amyloid peptide is believed to cause capillary rupture and atrophy, leaving free β-amyloid peptide cores in the surrounding brain parenchyma [130]. Thus, the predominant amount of β-amyloid peptide may come from the systemic circulation, moving to the brain parenchyma, or simply remaining in it after vascular atrophy, becoming the core of amyloid plaques [27,30,130]. This is supported by the observation that smaller vessels, such as capillaries, which are prone to rupture, show massive accumulation of β-amyloid peptide in the vessel wall [131]. Finally, over time, scars/cores with excess β-amyloid peptide are likely to develop into senile amyloid plaques [130].

After ischemia in animals and humans, microbleeding in the brain tissue has been described [96,97]. These small vascular hemorrhages recruit and activate platelets in the vicinity of the rupture. Post-ischemic neurodegenerative changes are chronic, many of these small vascular injuries may develop during survival after insult. Upon activation of platelets at the site of vasculitis, they release biologically active molecules that can modify the function of the vascular wall [123,132]. It should be noted that platelets contain very large amounts of the amyloid protein precursor that generates amyloid as well as amyloid itself. A smaller, 40 amino acid form of amyloid predominates in platelets [123] and it has been suggested that this peptide, as with in AD cerebral amyloid angiopathy, may accumulate and contribute to thrombus formation at the site of injury. It is now proposed that these accumulations of circulatory and platelet-derived amyloid in the damaged vascular wall in the early post-ischemia period may be an inducer of cerebral amyloid angiopathy. This phenomenon can act as a vicious circle after an injury.

## 9. The Role of the Ischemic BBB in the Maturation of AD

In contrast to the classical view of AD, current data indicate that ischemia-reperfusion injury contributes to progressing neurodegeneration in AD (Figure 1 and Figure 2) [9,18,20,31,37,38,39,41,46,79,128,129,133,134]. We propose that ischemic BBB acts as a maturation phenomenon in AD—it means the dysfunctional permeability and defective clearance of β-amyloid peptide across the ischemic BBB cause amyloid accumulation and maturation as amyloid plaques in AD (Figure 1 and Figure 2). This would result in premature senescence of the microcirculatory system and deficient angiogenesis of BBB vessels, regression of the neurovascular system, and neuroinflammation [65,135,136,137]. Finally, repeated ischemic insults with repeated BBB dysfunction are observed. Dysfunction in BBB cells within the microvascular system could probably disrupt this system and begin the neuropathogenic disease cascade associated with abnormal β-amyloid peptide metabolism (Figure 2) [74,138]. A theoretical scheme that fits very well with the ischemia basis of AD etiology is proposed. We suggest that AD would start developing provided that at last two neuropathological events are present: brain ischemia and subsequent damage to BBB. These events inevitably lead to two main pathologies: brain ischemia is mainly associated with neuronal death e.g., in the hippocampus, ischemic and chronic BBB deficiency inducing final amyloid pathology (Figure 1 and Figure 2). Much evidence indicates that the neuropathogenesis of AD is related to neurovascular-ischemic pathology. With the help of this proposal, some novel therapeutic approaches may be suggested that could be implemented in AD to preclude permeability of ischemic BBB for β-amyloid peptide [42] and raise reverse β-amyloid peptide clearance from the brain after ischemia-reperfusion injury [29,30].

## 10. Conclusions

The brain has restricted almost the same answer to different pathogens. The evidence presented in this review points to similar neuropathological features of cerebral ischemia and AD [18,20,37,38,39,41,74,129,133,134,139]. Post-ischemic brain injury is a progressive neurodegenerative process affecting cognition, behavior, and overall functioning [12,100,101,102,103,104,140]. This makes neuropathogenesis and the relationship between post-ischemic dementia and AD widely discussed [18,20,37,38,39,41,74,75,125,133,134,141]. The significance of cerebral ischemia itself and the post-ischemic BBB in the pathophysiology of Alzheimer’s disease is now of more importance than previously thought. It is now accepted that post-ischemic dementia and dementia associated with AD share common mechanisms and risk factors [12,18,20,37,38,39,41,104,128,129]. This concept is supported by the fact that ischemia triggers amyloidogenic metabolism of the amyloid protein precursor and dysfunction of the tau protein [9,18,20,26,37,38,39,74,129]. In this review, we suggested that the β-amyloid peptide probably rapidly seals the lesion in the cerebral vessels initially post-ischemia [45] to limit the neuropathology associated with microbleeding, cerebral edema, and repeated microinfarcts [14,94,96]. Since β-amyloid peptide is continuously produced in the blood under both normal and pathological conditions [66,70,72,73], we suggest that β-amyloid peptide in the blood acts as a seal [45] to stop the influx of serum β-amyloid peptide into brain parenchyma during the primary opening of the BBB [121,122]. At present, it is extremely difficult to distinguish the physiological accumulation of β-amyloid peptide from its pathological deposition in the brain. However, current data indicate that in the chronic pathological state with very long-term recirculation after brain ischemia, the properties of amyloid are neuropathological. Age-related loss of vascular integrity and function must also be considered in this situation [109,110,142]. Abnormal BBB sealing possibly caused by the breakdown of blood purification from the β-amyloid peptide, thickening of the vascular wall, and transformation of amyloid protein precursor metabolism from non-amyloidogenic to amyloidogenic, leads to the development of cerebral amyloid angiopathy in association with the age of the patient [12,104,143]. Another line of evidence suggests that the massive deposition of β-amyloid peptide in the vascular wall of the brain leads in a vicious circle to progressive damage to the cerebral microvessels that represent the BBB, and this causes permanent damage of the BBB after ischemia [34,76]. These observations indicate that durable damage of the BBB with progressive and massive accumulation of β-amyloid peptide in the brain tissue after ischemia should be believed to be the ultimate element that irreversibly exacerbates the outcome of the post-ischemic brain with the development of dementia with the AD phenotype [12,100,101,102,103,104,140]. These data provide evidence that a common environmental event, such as brain ischemia, can generate the same molecular neuropathology as in AD [6,18,20,26,37,38,39,40,41,74,129,133,134,139]. In addition, these studies have shown that post-ischemic changes that mimic the neuropathological and neurochemical alterations in AD also induce the massive production of the β-amyloid peptide characteristic of AD [9,67,69,71,74,130]. Taken together, these data strongly indicate that the expression and accumulation of the β-amyloid peptide occurs concomitantly with neuronal cell death [115,144].

Advances in the study of the BBB are important from the point of view of understanding the role of BBB insufficiency in post-ischemic and AD dementia. The data indicate that by control of blood-to-brain and brain-to-blood passage of β-amyloid peptide, the BBB can self-limit β-amyloid peptide-mediated vascular injury and prevent/decrease the death of ischemic neurons after ischemia [27,29,30]. The post-ischemic permeability of the BBB to the platelets and the β-amyloid peptide itself causes cerebral amyloid angiopathy which consequently narrows or occludes the vessels, and this induces repeated additional ischemia of the adjacent brain parenchyma [12,112,123]. It seems that the blood-derived β-amyloid peptide is likely to play a main role in the accumulation of amyloid in cerebral vessels and brain parenchyma, provided that the integrity of the BBB is compromised by ischemic damage [70,72,73]. These observations indicate that the β-amyloid peptide accumulating in the ischemic brain and vessels, mainly of blood origin, collects focally as in AD. Changes in brain vessels, including degeneration of pericytes and astrocytes [13,50,53,118], thickening of the basement membrane, and changes in the shape of endothelial cells along with the accumulation and aggregation of platelets in their lumen occur mainly in the branching and bifurcations of vessels [10], and consequently lead to a local no-reflow phenomenon (Figure 1) [88]. Under these conditions, the removal of waste products from the damaged brain parenchyma into circulation is downright impossible. These data suggest that chronic ischemic insufficiency of the BBB leads to progressive β-amyloid peptide-dependent damage of ischemic neurons, which is a major contributor to the development of post-ischemic dementia with the AD phenotype.

Data analysis shows that one of the most important pathological changes in all stages after cerebral ischemia is a dramatic increase in the permeability of the BBB, which in turn leads to serious and irreversible consequences for patients [18,20,26,38,39,41]. We indicate that ischemic permeability of the BBB with chronic leakage of serum β-amyloid peptide into brain parenchyma leads to chronic and β-amyloid peptide-dependent death of ischemic neurons. Ischemia with recirculation probably increases the vulnerability of ischemic neuronal, glial, and endothelial cells to the toxic β-amyloid peptide. Furthermore, we suggest that progressive neurodegeneration after cerebral ischemia may be accelerated by premature aging of cerebral vessels and brain parenchyma in the absence of treatment [142]. As a result of ischemic stroke in humans, 120 million neurons, 830 billion synapses, and 714 km of myelinated fibers are lost every hour. Compared to the normal rate of neuron loss in an aging brain, an ischemic brain without treatment ages 3.6 years per hour [142].

Cerebral microvasculature has received a lot of interest in many brain diseases in recent years, such as ischemic stroke, AD, and aging, as it is a key element in maintaining or improving the vascular network of the brain and therefore its functioning [145]. In these diseases, the vascular network is damaged, mainly by damage to endothelial cells, with consequent lack of neurovascular effects due to lack of signaling, neural stem cell proliferation/migration, as well as lack of oxygen and nutrients normally supplied by endothelium and BBB exchange [138]. All of this leads to a cascade of events that contribute to the loss of cognition or other basic brain functions. Given that such injuries involve disruption of the BBB, which alters not only the homeostatic movement of ions, nutrients, and cells between the blood and the brain but also the clearance of neurotoxic substances [145]. Impairment of clearance pathways may also promote the accumulation of unwanted harmful proteinopathic molecules such as amyloid and tau protein [145]. Damage to the vascular network can also lead to a reduction in cerebral blood flow, which results in reduced oxygenation of the brain and thus induces hypoxia of the microenvironment [138,145]. On the other hand, fewer trophic factors are produced and released in response to injury, which consequently reduces chemotactic signals for endothelial cell migration and proliferation, contributing to greater sensitivity and vulnerability of neurons and glial cells to ischemia [138,145]. In addition, damaged parts of the vascular network may eventually degenerate [57], leading to a reduction in the vascular network, which exacerbates all the above-mentioned problems [138,145].

In summary, the neuropathogenesis of ischemia-associated dementia involves first-stage direct ischemic death of neuronal cells (necrosis, apoptosis). In the second stage, acute and chronic ischemic dysfunction of the BBB with leakage of the β-amyloid peptide from the plasma into the brain parenchyma triggers additional death of ischemic neurons dependent on amyloid and finally develops amyloid plaques and ends in full-blown dementia with the AD phenotype [12,18,20,26,39,41,74,104,130]. It appears that AD may result from repeated sub-lethal and silent ischemic brain episodes (microstrokes) that attack and slowly steal the minds of their victims. At the clinical level, such detailed vascular analysis is still not possible: instead, current imaging systems can analyze cerebral blood flow. Nevertheless, this represents a promising possibility in combination with potential pro-angiogenic therapies.

## 11. Current Message from the Lab for the Clinic

Vascular factors contributing to the development of AD are attracting increasing interest from a research and clinical perspective, while clinical implementation remains limited due to the limited number of studies, but allows lifestyle changes to be recommended to reduce vascular risk factors leading to cerebral ischemia. Due to this limitation, it is extremely important to continue research in the search for specific biomarkers of vascular changes in patients with AD, especially before the onset of cognitive impairment. For example, they can help identify people who are asymptomatic long before the onset of AD and who have a family history of AD. Future efforts should focus on finding those biomarkers that would be most sensitive and specific for the above purposes. Meanwhile, clinicians have several options for advising patients on ways to alleviate vascular disease and potentially AD (mental and physical activity, proper diet, regulation of blood pressure and level of lipids, dietary regimen, caloric restriction, etc.). Although the exact nature and explanation of the vascular–ischemic relationship and AD remain unclear, vascular dysregulation (ischemia, hypoperfusion) and AD are associated pathophysiologically and neuropathologically, sharing imaging findings and risk factors. Several pathways, including microvascular infarcts and dysfunction of the ischemic BBB, appear to contribute to the vascular dysfunction associated with the development of AD. Identification of sensitive vascular biomarkers is a priority in the search for patients at risk of cognitive decline and dementia. Chronic cerebral hypoperfusion, ischemia, repeated ischemic episodes, and other episodes of cerebrovascular disorders show promising results, but more evidence needs to be gathered before any definitive conclusions can be drawn. Targeted longitudinal studies evaluating specific vascular markers in populations at risk of AD are essential to elucidate the complex relationship between vascular diseases such as ischemia with dementia, more specifically AD. Focusing on the ischemic factors contributing to the development of AD through their tight control is an attractive and potentially useful way to prevent AD and/or its outcomes. Thus, the current evidence supports the control of vascular factors leading to global or focal ischemia as a potential first step in preventing AD-related changes and could be further implemented in a broader clinical system aimed at slowing the progression of AD.

## Figures and Tables

**Figure 1 ijms-24-10739-f001:**
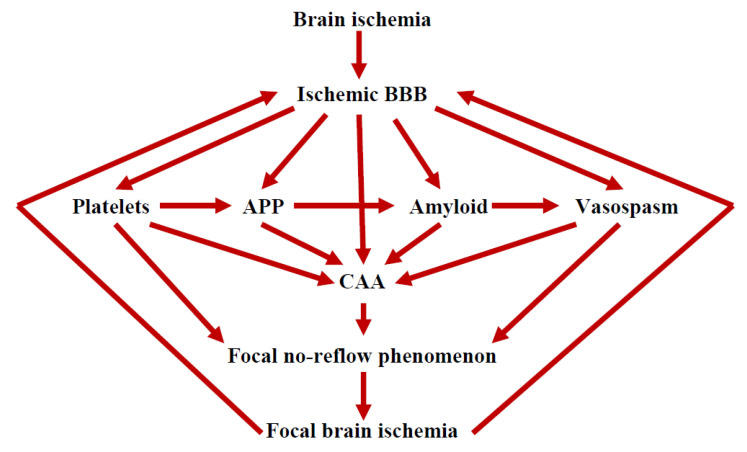
Hypothetical illustration of the role of ischemic brain injury and its relationship with the ischemic blood–brain barrier (BBB) in generating Alzheimer’s disease-type neuropathology in a vicious circle. Cerebral ischemia leads to increased permeability of the BBB, which drives platelets, an amyloid protein precursor (APP), and amyloid to various structures in the brain, leading to the development of cerebral amyloid angiopathy (CAA), which results in reduced cerebral blood flow. which triggers repeated focal cerebral ischemia with increased permeability of the BBB in a vicious circle.

**Figure 2 ijms-24-10739-f002:**
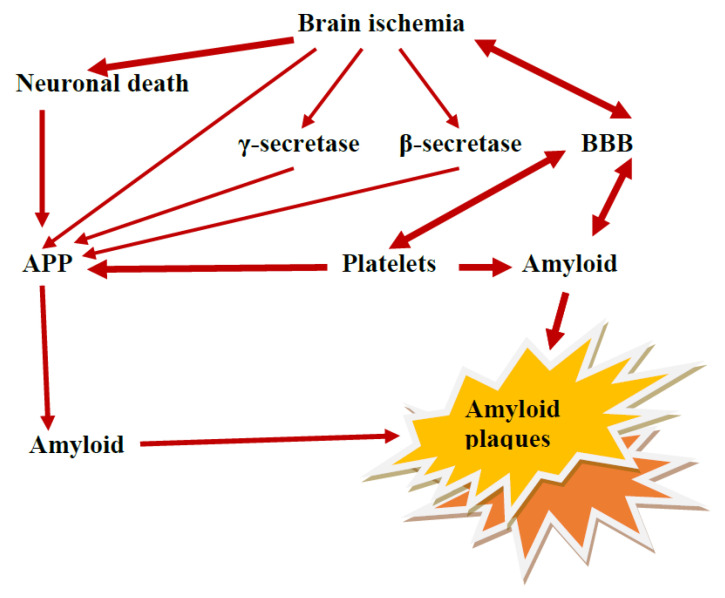
Two pathways of β-amyloid peptide formation and accumulation after ischemic brain injury and their impact on the development of amyloid plaques. Thickness of response intensity—arrows; BBB—blood–brain barrier; APP—amyloid protein precursor.

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
