# Peer review of "Post-Ischemic Permeability of the Blood–Brain Barrier to Amyloid and Platelets as a Factor in the Maturation of Alzheimer’s Disease-Type Brain Neurodegeneration"

_ijms, 2023, doi:10.3390/ijms241310739_

Round 1
Reviewer 1 Report
The Review focuses on post-ischemic impact on BBB permeability as related to AD associated neurodegeneration. Points of concern, discrepancy, and/or required clarification are as follows:
· Title: the “Trojan horse” metaphor has been used heavily in drug development referring to enhance permeability as it relates to purposeful strategies. Suggest a rework of the title without the term and with a more focused form that can be fully grasped at a single quick reading as to encompassing review objectives (i.e. ischemia, BBB permeability, AD impact) without unique terminology.
· Section-1 introduction paragraphs need rework. The first paragraph particularly has several redundancies in concept and very generalized information. The first paragraph may simply be better place (in abbreviated form) in section 3 (…Structure and functions of the BBB). Suggest introduction open up with information on AD and Aβ, transition to BBB definition/concept and thus lead into second paragraph.
· “uncellular” is inappropriate term (line 146, 187), replace with “non-cellular”
· Lines 207, 210, use term “above” in reference to impacted areas, and should be replaced with appropriately descriptive term(s).
· Further delineate meaning of the “failure” (line 270), is this actually meaning “paracellular permeability”?
· Over repeating the insertion of “(figure-1)” and “(figure-2)” in the review, where it is not completely necessary for the respective point(s) addressed.
· The arrow design figure-2 lacks a concise and/or informative form. At least one high-quality graphic-figure of the endothelium with associated activating-effector components of ischemia as related to Aβ and AD related changes (relatively inclusive of overall theme of review) would be of significant added value for the publication.
· Line 482-484 note common risk factors. Would be ideal to have figure/table of those factors across a dynamic timeline.
· Conclusion section is both redundant and scattered, requiring a focused edit and likely moving some information to different sections.
· Minor points:
o suggest abbreviating blood-brain barrier (BBB) and Alzheimer’s disease (AD) throughout the manuscript.
o Suggestion: Eliminate “We further investigate the role of the brain ischemic episode with” (line 279) and simply start the sentence with “An alternative theory…”
o Author Contributions: Looks like authors merely cut-n-pasted the instructions for this section rather than actually filling it out appropriately.
Summary:
Overall, the review has insightful and valuable information but is dyssynchronous in form, which makes it distracting to read. Conclusion section needs significant rework. Figures are of low quality and could be greatly enhanced. Inclusion of a timeline-based figure would greatly help review.
The quality of English is sound. Editing relative to placement of information and reduction of redundant information needed.
Author Response
Comments and Suggestions for Authors
The Review focuses on post-ischemic impact on BBB permeability as related to AD associated neurodegeneration. Points of concern, discrepancy, and/or required clarification are as follows:
- Title: the “Trojan horse” metaphor has been used heavily in drug development referring to enhance permeability as it relates to purposeful strategies. Suggest a rework of the title without the term and with a more focused form that can be fully grasped at a single quick reading as to encompassing review objectives (i.e. ischemia, BBB permeability, AD impact) without unique terminology.
Authors: Thank you for this suggestion. The title was changed and “Trojan horse” is no more present.
- Section-1 introduction paragraphs need rework. The first paragraph particularly has several redundancies in concept and very generalized information. The first paragraph may simply be better place (in abbreviated form) in section 3 (…Structure and functions of the BBB). Suggest introduction open up with information on AD and Aβ, transition to BBB definition/concept and thus lead into second paragraph.
Authors: The first para was transferred to section 3 as suggested and the reference numbers were changed accordingly.
- “uncellular” is inappropriate term (line 146, 187), replace with “non-cellular”
Authors: “Uncellular” was replaced with “non-cellular”.
- Lines 207, 210, use term “above” in reference to impacted areas, and should be replaced with appropriately descriptive term(s).
Authors: Done (lines 205 and 208).
- Further delineate meaning of the “failure” (line 270), is this actually meaning “paracellular permeability”?
Authors: We change “failure” for damage.
- Over repeating the insertion of “(figure-1)” and “(figure-2)” in the review, where it is not completely necessary for the respective point(s) addressed.
Authors: The number of figure insertions was considerably reduced.
- The arrow design figure-2 lacks a concise and/or informative form. At least one high-quality graphic-figure of the endothelium with associated activating-effector components of ischemia as related to Aβ and AD related changes (relatively inclusive of overall theme of review) would be of significant added value for the publication.
- Line 482-484 note common risk factors. Would be ideal to have figure/table of those factors across a dynamic timeline.
Authors: Thank you for these remarks on figures. However, in our opinion the present figures are quite sufficient and summarizing the whole problem.
- Conclusion section is both redundant and scattered, requiring a focused edit and likely moving some information to different sections.
Authors: Conclusions were considerably shortened which was associated with a reduction in the reference number. Now the total number of references is 169.
- Minor points:
- suggest abbreviating blood-brain barrier (BBB) and Alzheimer’s disease (AD) throughout the manuscript.
- Authors: Done
- Suggestion: Eliminate “We further investigate the role of the brain ischemic episode with” (line 279) and simply start the sentence with “An alternative theory…”
- Authors: The sentence was modified (line 276).
Author Contributions: Looks like authors merely cut-n-pasted the instructions for this section rather than actually filling it out appropriately.
Authors: The editorial instructions were deleted.
Summary:
Overall, the review has insightful and valuable information but is dyssynchronous in form, which makes it distracting to read. Conclusion section needs significant rework. Figures are of low quality and could be greatly enhanced. Inclusion of a timeline-based figure would greatly help review.
Authors: Conclusions reworked. As for the quality of the figures, we have a different opinion. We find them concise and easy to understand for the casual reader.
Changes in yellow.
Reviewer 2 Report
The clarity and structure of your paper are commendable, as you successfully navigate through complex concepts and present your findings in a coherent manner. Your attention to detail in providing comprehensive explanations and referencing relevant literature showcases your thorough understanding of the subject matter.
Author Response
Comments and Suggestions for Authors
The clarity and structure of your paper are commendable, as you successfully navigate through complex concepts and present your findings in a coherent manner. Your attention to detail in providing comprehensive explanations and referencing relevant literature showcases your thorough understanding of the subject matter.
Authors: We deeply appreciate your positive comments.
General: All changes (including the editorial comments) in the text were highlighted in yellow.
Round 2
Reviewer 1 Report
- First 7 pages still fully spell out "blood-brain barrier" instead of abbreviating "BBB".
- would suggest changing "maturation" with "development".
Language is clear.
Author Response
Answers to the reviewer remarks:
- First 7 pages still fully spell out "blood-brain barrier" instead of abbreviating "BBB".
Authors: We apologize for the incomplete revision in this regard. Now, the abbreviation of BBB is present in the entire text.
- - would suggest changing "maturation" with "development".
Authors: We would, however, insist on leaving the term “maturation”, which fully reflects what we actually mean.